# A Study on the Network Effectiveness of Sustainable K-Fashion and Beauty Creator Media (Social Media) in the Digital Era

**Younkue Na** [1], **Sungmin Kang** [2,*] and **Hyeyeon Jeong** [3]

1   Department of Art & Culture Research Institute, Chung-Ang University, 84 Heukseok-ro, Dongjak-gu, Seoul 06974, Korea; nyk901@gmail.com
2   College of Business and Economics, Chung-Ang University, 84 Heukseok-ro, Dongjak-gu, Seoul 06974, Korea
3   Department of Fashion Business Management, Fashion Institute of Technology (FIT), State University of New York (SUNY) Korea, 119 Songdo Moonhwa-ro, Yeonsu-gu, Incheon 21985, Korea; hyeyeon_jeong@fitnyc.edu
*   Correspondence: smkang@cau.ac.kr; Tel.: +82-10-2243-8647

**Abstract:** With the convergence of various media in the digital era, the influence of Korean fashion/beauty on popular culture is growing rapidly. This study examines the sustainable relationship between the content and community characteristics of Korean fashion/beauty creator media, the associated social exchange relationships, and the effectiveness of the network among international consumers. In total, 614 international consumers who had made Korean fashion product purchases, viewed Korean fashion creator media, and shared information related to Korean fashion at least once were selected as a sample. Frequency analysis, reliability and validity analysis, measurement model analysis, and path analysis were conducted using SPSS and AMOS. The results showed that, first, content uniqueness had a significant effect on perceived similarity, although content continuity did not. In addition, content uniqueness and content continuity both had a significant effect on emotional expectations. Second, community scalability and community cohesion both had a significant effect on perceived similarity, and community scalability and community cohesion had a significant effect on emotional expectations. Third, perceived similarity had a significant effect on both emotional expectation consciousness and parasocial interaction, and emotional expectation consciousness had a significant effect on parasocial interaction. Finally, parasocial interaction had a significant effect on fad-like behavior. Through this, this study expanded the scope of academic research by linking the contents and community characteristics of Korean fashion/beauty creator media with research problems in the field of social exchange from the perspective of network effectiveness. Integrating this with the existing studies on consumer acceptance of Hallyu culture is expected to lead to the development of a more descriptive theoretical model for the formation of attitudes and purchase intentions toward Korean fashion/beauty products.

**Keywords:** fashion/beauty creator media; content characteristics; community characteristics; social exchange relationship; network effectiveness

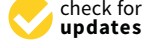

## 1. Introduction

With the development of the Internet and the global popularity of devices such as smartphones, users are increasingly finding information through media such as portal sites and social networking sites (SNS) rather than traditional media. Among them, creator media content, which centers around the participatory consumer, is widely replicated, produced, and consumed, playing a significant role in allowing new cultural content to be easily delivered to the global public in the digital era [1]. This creator media represents a new social media commerce format that connects to sustainable commerce through video content such as product introductions, user guides, and reviews via individual creative broadcasting platforms such as YouTube and multi-channel networks (MCNs). Creator media is developing to support marketing campaigns or share profits by introducing products or generating business, such as direct buying (home shopping, open market, social

commerce, etc.). Moreover, with today's global industrialization of social media culture, everyone can share the same culture within a short time, thereby rapidly popularizing cultural homogenization, cultural practice, and enjoyment. The same cultural code, formed in this way, is the basis for social integration through smooth communication.

In particular, the cultural environment of creator media (social media) formed the background for the creation of the sustainable Hallyu (literally, "Korean Wave"—mainly referring to the phenomenon of Korean popular culture trends abroad [2]) phenomenon, which is based on Korean popular culture [3]. Accordingly, MCN operators related to Korean creator media are actively advancing overseas, establishing a bridgehead for foreign expansion. Among them, the fashion/beauty content area has allowed it to easily enter major overseas markets thanks to the popularity of Korean culture in terms of Hallyu, Korean cosmetics, and Korean fashion brands. In addition, the number of global Hallyu fans, who enjoy Korean content such as Korean drama and K-pop, is increasing. They are actively learning about Korean culture, for example, by buying cosmetics and Korean products and following the makeup and fashion of their favorite stars [4]. As attempts to increase overseas sales are intensifying in the fashion and beauty field (PPL, advertising-linked sales, etc.), Korean creator media plays an important role by pursuing active collaboration between content and products.

In addition, in the big picture of Hallyu, creator media is creating a strategic environment where fashion/beauty creators can share the flow of sustainable Hallyu mass culture through K-drama, K-pop, K-culture, and K-media content. Consequently, many fashion/beauty creators are popularizing Korean-style fashion, style, makeup, skincare know-how, and other fashion/beauty information, and the proportion of views of the content produced by Korean fashion/beauty creators is increasing [5]. Products recommended by creators gain considerable attention online, sell out quickly, or show significant marketing and cultural ripple effects; therefore, it is a good time to have additional effects of distribution activation based on creators' expertise. Given this background, the strategic use of creator media to distribute content produced by individual Korean fashion/beauty creators with outstanding content production capabilities and *ki* (talent) can create an opportunity to further activate and sustain the Hallyu boom and its overseas market development.

However, creator media is still in its early stages, and related academic research is lacking. Creator media is an open online tool and media platform used to share thoughts, opinions, experiences, and perspectives among users as a form of social media, indicating that mutual sharing is an important factor [6]. This type of media creates an environment that is the center of communication activities throughout the network. Accordingly, the multimedia content and communities of creator media are produced with various intentions by their creators and by user networks in multiple fields; further, they are intended to be consumed as a network according to the creative direction of the content [7]. Nevertheless, related prior studies have focused only on the platform structure in which the creator's content is published, while user networks have been excluded [8–10]. Moreover, there has been no attempt, thus far, to define the value creation of social networks of fashion/beauty related creator media from the user's perspective, and more in-depth consideration is needed to demonstrate the implications of their characteristics and effects.

Therefore, in this study, the content and community characteristics of Korean fashion/beauty creator media are presented in detail, including sub-variables for each dimension. In addition, the characteristics of creator media are evaluated in terms of cognitive, emotional, and behavioral factors to measure their communication effectiveness. Specifically, the structural influences of the content and community characteristics of Korean creator media on the social exchange relationships between this media and its participants, as well as the network effect of the participant group, are verified. This study is expected to increase the possibility of academic and practical application through the strategic evaluation of the sustainable social network's role and performance between Korean fashion/beauty creator media and international consumers.

## 2. Theoretical Background

### 2.1. Creator Media Content Characteristics

Creator media platforms, in which high-level content that is discovered and produced by individual creators is distributed, has characteristics that transcend time and location. It is increasingly possible to effectively target advertisements through detailed content provision and user analysis, making the creator media platform a new Hallyu distribution channel. Moreover, fashion/beauty content creators have comprised one of the most active fields from the early days of individual media creation and have emerged as a new industrial market that offers various business possibilities. Content creativity places more importance on "what" is delivered than "how" the message is delivered. According to one previous study, the psychological mechanism that causes content creativity to elicit a consumer response is based on originality, brand relevance, completeness and composition, and consumer empathy [11]. The same study defined content user satisfaction factors in terms of satisfaction with the content itself, the system (tools and environment), and the expected value (i.e., the degree of agreement between expectations and results). Such content can fulfill its mission by assuming a role as a core platform of the creative knowledge-based economy, not merely as entertainment content. Moreover, the main goal of creator media is to enable communication by achieving more efficient exposure to consumers through the use of creative content. To this end, Choi [12] stated that media-centric characteristics such as openness, dialogue, participation, and connection, which traditional media do not possess, are consistent with media creative characteristics such as participation, play, interaction, and dissemination. This is said to be an important core content of the current media strategy for communication. Accordingly, it is necessary to understand the characteristics of the fashion/beauty creator media content that can create and seek differentiated content value through consumer perception, which is considered to be a strategic element. Moreover, another content characteristic of creator media, which is closely related to overseas fashion/beauty interested consumers, will increase its use as a communication tool.

### 2.2. Creator Media Community Characteristics

In the era of participatory mass media, sharing and openness are evolving in the form of micro media and personal media. In addition, the 1.0 paradigm of content production by a minority and consumption by the majority has disappeared, making way for a 2.0 paradigm aimed at participatory production and consumption [13]. User generated content (UGC) is emerging as an important way for users to creatively share network spaces such as creator media in this open and participatory web environment. In addition, the rise of a community within the major creator media, which becomes a user's "place of presentation," is emerging as a new cultural trend. In this regard, Nam and Kim [6] reported that media expansion, service expansion, playful media consumption, content re-diffusion, consumer participation and sharing, change in media subjects, communication using collective intelligence, escaping the temporal and spatial constraints of existing media, and the creation of a completely new media and content culture are the characteristics of the present time when media is used more creatively. As such, creator media users are able to spread individual values and enable information acquisition, communication, community, and small culture formation through social media that is more personal and expandable. Further, creator media users are centered around their relationships with other users and want to develop important resources, a sense of belonging, and awareness sharing through the community [14]. This sense of membership is created by providing emotional safety to members and is the basis for each person expressing their desires and feelings and forming a sense of intimacy with others, sharing a common purpose, activity, oral tradition, and common values [15]. Moreover, creator media is evolving toward emphasizing communication and relationship formation with others through openness and scalability as communication tools between users [16]. In this regard, Preece et al. [17] suggested shared purpose (title or aim, additional information), members (access, division

of roles, effective communication), and policies (registration, operation, trust, security) as a guideline for social relationship planning in the framework of participatory community-centered development. Examining a community that shares opinions and relationships through creator media offers an opportunity to explore new communication directions by more precisely revealing the behavior of foreign users interested in Korean fashion/beauty.

### 2.3. Social Exchange Relationship

In the accumulation and exchange of knowledge within a group, the importance of social relationships between members, who are the key subjects, is increasing [18]. Meanwhile, in the expansion of content from the individual level of one-person creator media to meet the competitiveness of the organizational level, interaction between members and mutual cooperation and exchange in social networks can lead to enhanced individual and organizational performance. In this regard, Chung and Jung [19] found that social media such as creator media can be viewed as a means of interaction, social capital, and ties with others in that they can form a close psychological bond with other people. As such, members who are sensitive to the latest trends and information and have a strong sense of collective sharing are enhancing their communication skills by voluntarily participating in the social network space within creator media and sharing knowledge and experiences with each other. In particular, the value of using this communication ability is an economically shared value that reproduces new knowledge and creates sympathetic energy through cooperation in the community formed by members. Therefore, it is necessary to maintain and develop it so that a sustainable sympathy and exchange of information occurs in the implementation process [20].

Ultimately, the more the group is cohesive, the more likely its members will adopt similar attitudes, values, or behavior patterns; if members have similar values, attitudes, or cultural backgrounds, the group is more likely to be aggregated [21]. In addition, since groups with strong group cohesion form a solid social identity that can increase the desire to help each other and produce a sense of expectation among the members, the cohesion of the group becomes mutually cooperative and job-related. This is recognized as an important situational factor in organizational actions (i.e., social exchange relationships) that can contribute to performance improvement [22]. As such, there have been various attempts to define the exchange relationship within a group [23]. The structural characteristics of social exchange relationships, such as similarity and expectations, may play an important mediating role between the characteristics of creator media and its performance.

### 2.4. Network Effectiveness

As a communication tool between users, social media is evolving in the direction of emphasizing relationship formation and communication with others through extensibility and openness [24]. In this regard, it is said that successful communities share a common purpose, activity, and common oral traditions and values [17], and above all, in the process of making decisions by imitating information from other nodes, it is said that information spreads rapidly and this causes information contagious effects [25].

When using the communication effect model to evaluate creator media characteristics with a focus on the reactive behaviors of the audience [26], the public's evaluation of creator media content/communities should be comprehensively considered in cognitive, emotional, and behavioral terms. Park and Chae [27] suggested several rules between the two exchange parties in a social exchange, such as reciprocity, negotiation, relationship, altruism, common benefits, identity, consistency, and competition principles. Social exchange in social media emphasizes the importance of cognitive and emotional aspects; in particular, the overall emotion generated by social exchange triggers cognitive efforts (more specific emotions, bonds to social objects, and various outcomes through this attribution process) to understand the cause. In the case of a joint task related to social exchange, explanations can be made based on social units (relationships, networks, groups), and such assumptions suggest the role and meaning of emotions in the process of social exchange. In addition,

Suh and Kim [28] stated that when the emotional bond with an online brand community increases, identification with the community also increases, and that this similarity and emotional bond are the result of interactions between members or between members and community operators.

Social interaction is critical for forming social similarities, shared values, and cooperative/individual interrelationships as social capital at the relational level. In this regard, social interaction encourages other related preferences such as altruism and fairness; since these social relationships have inherently similar preferences, the acceptance of individuals in the network is influenced by the acceptance of those who are connected to them [29]. Moreover, if a provider recommends attractive products in their content, it can increase the efficiency of information retrieval for consumers, and people with similar interests or inclinations are more likely to purchase similar products in a similar environment [30]. As such, in the behavioral stage, the behavior of members in the social network becomes information and influences the behavior of their neighbors, which produces the phenomenon of fads [31].

### 3. Research Method and Procedure

#### 3.1. Research Model and Hypotheses

Figure 1 presents the present study's research model. The research hypotheses are further discussed below, examining the causal relationships involved in creator media based on the findings of previous research.

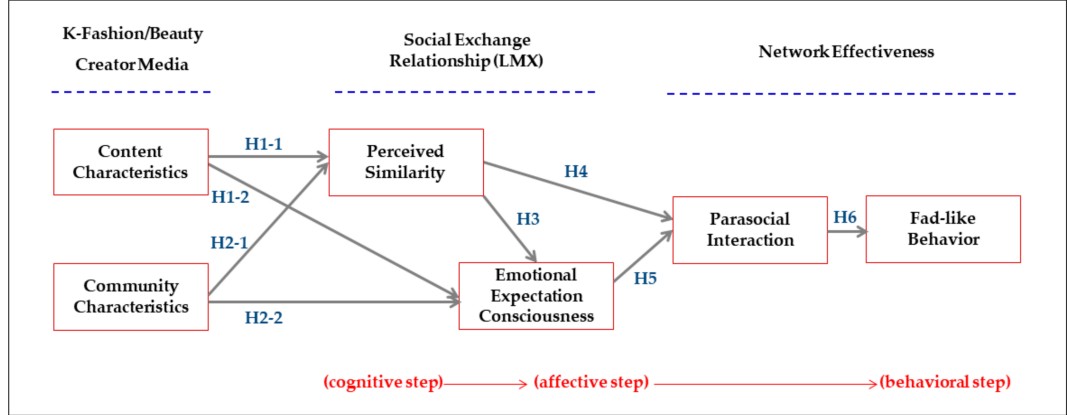

**Figure 1.** Research model.

3.1.1. Relationship between Content and Community Characteristics of Korean Fashion/Beauty Creator Media and Social Exchange

The main goal of creator media is to achieve more efficient exposure to consumers through the use of creative content that facilitates communication. Choi [12] suggested that the characteristics of traditional media, such as participation, openness, dialogue, and connection, do not have, at their core, a strategy for effective communication, which is more consistent with the creator media characteristics of participation, dissemination, interaction, and play. Specifically, Noh [32] defined the characteristics of media creativity as consumer participation, dissemination, interaction, play and media discrimination, complementation, connection and original creativity, innovation, and storytelling. Further, Nam and Kim [6] demonstrated that service expansion, media expansion, content redistribution, playful media consumption, media subject change, consumer participation and sharing, communication using collective intelligence, creation of completely new media, case verification, escape from the temporal and spatial constraints of traditional media, and the creation of a content culture are the characteristics of creative social media.

Social media, including creator media, can be viewed as a means to interact with others and develop social capital and bonds as it can produce close psychological ties and the

possibility of communication with others [19]. As such, members who are sensitive to the latest trends and information and have a strong sense of collectivity voluntarily participate in the social network space of creator media, sharing knowledge and experiences with each other and building a solid social exchange relationship. This represents a shared economic value that reproduces new knowledge and creates empathy through cooperation in the community that is formed by its members. It is necessary to maintain and develop sustainable empathy and information exchange in the process of implementation [20]. Individuals can deliver their opinions and perspectives through creator media to specific or unspecified people by gathering information on the subject of interest and posting their opinions [33]. As such, creator media users have a clear desire for content and community, carefully consider the information received through this community, and strongly express reasonable intentions to select the relevant information within a social exchange relationship. The characteristics of creator media content and the community sharing this information affect the social exchange relationship with and among users; thus, the following research hypotheses were proposed:

**Hypothesis 1 (H1).** *The content characteristics of Korean fashion/beauty creator media will have a significant influence on the social exchange relationship (perceived similarity, emotional expectation consciousness) between the media and community participants.*

**Hypothesis 2 (H2).** *The community characteristics of Korean fashion/beauty creator media will have a significant influence on the social exchange relationship (perceived similarity, emotional expectation consciousness) between the media and community participants.*

3.1.2. Relationship between Social Exchange Relationships and Network Effectiveness in Korean Fashion/Beauty Creator Media

The formation of a social network does not necessarily mean that information is easily shared, and this sharing may vary depending on the nature of the network formed among its members. Sin and Kim [34] stated that members with many similarities have more pleasant feelings toward each other and communicate easily by sharing interests with their peers and having expectations from them. From the standpoint of uncertainty reduction theory, group cohesion increases when group members expect to communicate with more members to reduce uncertainty [35,36]. Furthermore, social exchange relationships reflect the meaning of the group level, which implies that members feel a degree of attraction and emotional attachment toward each other, the intensity of the group's residual consciousness, a sense of unity, a sense of bonding or solidarity, teamwork, loyalty and unity, and a collective spirit. A group's unity means that the members of the cohesive group are willing to tackle problems together [37]. In this way, in terms of social exchange theory, social relationships are the process of continuously exchanging information between members. Interdependence is formed since each party has the necessary resources and trusts that the other party will give them positive results [38].

Members of creator media communities exchange information and perform a group process while communicating through social networks and their social capital. Creator media, like social media, not only forms new relationships by facilitating communication through a follow-up extension method but also functions to maintain existing connections or cohesive social capital [39]. Moreover, to expand the individual-level knowledge to the competitive organizational level, interaction between members and mutual cooperation and exchange in social networks can lead to improved individual and organizational performance. This positively contributes to the process of accumulating and sharing knowledge and information to facilitate organizational problem solving [40]. Accordingly, social pressure from social media communication promotes bandwagon effects and results in herd behavior [41]. As such, members of groups with high parasocial interactions show high levels of morale and motivation, find their group attractive, and increase the intensity of their joint efforts, which is considered to have an effect on fad-like behavior. Thus, the following research hypotheses were proposed:

**Hypothesis 3 (H3).** *The perceived similarity between Korean fashion/beauty creator media and its participants will have a significant influence on emotional expectation consciousness.*

**Hypothesis 4 (H4).** *The perceived similarity between Korean fashion/beauty creator media and its participants will have a significant influence on the parasocial interaction of the participant group.*

**Hypothesis 5 (H5).** *The emotional expectation consciousness between Korean fashion/beauty creator media and its participants will have a significant influence on the parasocial interactions of the participant group.*

**Hypothesis 6 (H6).** *In Korean fashion/beauty creator media, the parasocial interaction of the participant group will have a significant influence on fad-like behavior.*

*3.2. Measurement Tools*

This study's measurement tool comprised questions on the content characteristics, community characteristics, perceived similarity, emotional expectation consciousness, parasocial interaction, fad-like behavior, and demographic characteristics found within Korean fashion/beauty creator media. Content characteristics were measured using six items in the dimensions of content uniqueness and content continuity, which assessed the uniqueness, retention, and continuity of Korean fashion/beauty creator media content. The measurement items were based on previous studies by Kim and Choi [3], Nam and Kim [6], Nam and Park [1], Lee [11], Lee and Lee [14], Korea Creative Content Agency [4], Pantiti [42], Preece et al. [17], etc. Community characteristics were measured using six items in the dimensions of community scalability and community cohesion, which are relational valuations that reinforce the solidarity between Korean fashion/beauty creator media services and their consumers. These measurement items were based on previous studies by Kim [22], Mok and Youm [43], Park [15], Choi [12], Ju and Han [13], Preece et al. [17], Wasko and Faraj [29], etc. Social exchange relationships were measured using six items in dimensions of perceived similarity and emotional expectation consciousness, which are the characteristics of the exchange relationship at the cognitive and emotional level. The measurement items were developed based on studies by Kim [33], Noh and Lee [35], Park [23], Park [21], You [20], You [18], Chung and Jung [19], Ma and Qu [44], Landry et al. [45], etc. Finally, network effectiveness was measured using six items in the dimensions of parasocial interaction and fad-like behavior, which represent network effectiveness at the behavioral stage among the Korean fashion/beauty creator media participant group. These measurement items were based on studies by Suh and Kim [28], Rhee et al. [26], Yu et al. [37], Han and Ok [31], Blight et al. [46], Ellison et al. [39], He et al. [30], Goldenberg et al. [41], Wasko and Faraj [29], etc.

*3.3. Data Collection and Analysis*

The data collection, investigation, and analysis methods of this study were as follows. First, to understand the content and community characteristics of Korean fashion/beauty creator media that influence the social exchange relationship and network effectiveness among fashion/beauty creators, focus group interviews (FGI) were conducted with creator media, Hallyu, and global fashion/beauty marketing experts, along with a literature review. To document the diverse and in-depth opinions of respondents before conducting a large-scale quantitative survey, open-ended questions about Korean fashion/beauty content as perceived by international consumers who had viewed Korean fashion/beauty creator media were then developed. Using the data derived from these steps and the frameworks of creator media from previous research, measurement items related to the content and the community characteristics perceived by the user in the creator media were developed. Further, considering the process of accepting the user's reaction to the use, relevant measurement items related to social exchange relationship (in terms of perceived similarity and emotional expectation consciousness) and network effectiveness (in terms of parasocial interaction and fad-like behavior) were developed.

Second, this study attempted to collect reliable and valid data to accurately evaluate the characteristics of Korean fashion/beauty creator media and produce measurement tools related to the social exchange relationships that influence the network effectiveness among foreign users. International consumers who were estimated to have a very high interest in Korean fashion/beauty creator media and were active in the Hallyu community were selected as the primary survey sample. Research subject involves creator media, including domestic live broadcasting platforms (e.g., Africa TV, Kakao TV, Naver TV, and Pandora TV), global platforms (e.g., YouTube, Facebook, Twitter, Tuchi TV, and Instagram), and Korean fashion/beauty content creator media provided by live broadcasts overseas (e.g., Dia TV, Treasure Hunter, Leferry, and Sandbox Network) on MCN platforms.

Cha et al. [47] suggested that the phenomenon of sharing and spreading in social networks should be considered as acceptance through social contagion and should be included in the phenomenon of social contagion when there is at least one neighbor (member) connected to the survey target who has shared the content owned by this target. Therefore, the present study carefully considered the setting of each research subject in order to collect the most reliable data. To this end, international consumers/users who had purchased Korean fashion/beauty products, viewed Korean fashion/beauty creator media, and shared information related to Korean fashion/beauty at least once (comments, scraps ["clipping"]) were selected as research participants. A random sampling method was adopted, and participants were approached through a link to a questionnaire that was posted to a Hallyu community bulletin board and on creator media platforms in Korea and abroad. A preliminary ("pretest") survey (1–15 October 2020; 50 people) and a main survey (November 1–10 December 2020; 630 people) were conducted; the final data comprised 614 responses with no missing data.

The frequency analysis of the general characteristics of the sample and internal consistency aspects of reliability and validity of were verified using SPSS ver. 23.0 and AMOS ver. 23.0. Further, measurement model analysis and path analysis were conducted to verify the structural influence relationship among the content and community characteristics of Korean fashion/beauty creator media, perceived similarity, emotional expectation consciousness, parasocial interaction, and fad-like behavior.

## 4. Empirical Analysis and Results

### 4.1. Demographic Characteristics of the Sample

In terms of gender, the sample included 89.1% (547) women and 10.9% (67) men. Overall, 38.8% (238) of the participants were in their 20s, 26.9% (165) in their 30s, 18.9% (116) in their teens, 11.1% (68) in their 40s, and 4.6% (28) in their 50s. In terms of education, 44% (272 respondents) of the participants had obtained university enrollment/graduation, 18.9% (116) had junior college enrollment/graduation, 18.9% (116) had graduate school enrollment or higher, and 17.9% (110) had high school graduation or less. Lastly, 38.1% (234) were office workers, 36.5% (224) were students, 5.7% (35) were housewives, 11.7% (72) were unemployed, and 8.0% (49) selected "Other."

### 4.2. Reliability and Validity Analysis

Before evaluating the measurement model, reliability was examined by calculating the confidence coefficient (Cronbach's α) to verify the internal consistency of each research concept. As a result of a factor analysis using Varimax rotation, for the six items that described the content characteristics of Korean fashion/beauty creator media (Table 1), the two factors of "content uniqueness" (three items) and "content continuity" (three items) were extracted with an eigenvalue of 1.000 or higher. The total variance explained by these two factors was 70.908%, and the Cronbach's α were all 0.814 or higher, indicating high reliability of the items.

**Table 1.** Reliability and validity of relevant variables in the research model.

| Variable | Measurement Item | Eigenvalue | Factor Loading | Variance | Cronbach's α |
|---|---|---|---|---|---|
| Content Uniqueness | - Content Purpose<br>- Practical Usefulness<br>- Content Trust | 2.168 | 0.833<br>0.745<br>0.721 | 39.089 | 0.841 |
| Content Continuity | - Content Uniqueness<br>- Program Diversity<br>- Feeling of Synesthesia | 1.964 | 0.879<br>0.748<br>0.641 | 31.819 | 0.814 |
| Community Scalability | - Internationalization<br>- Network Continuity<br>- Network Grouping | 2.636 | 0.899<br>0.854<br>0.851 | 39.285 | 0.915 |
| Community Cohesion | - Intimacy Structure<br>- Collaborative Partnership<br>- Trust-based Community | 2.519 | 0.857<br>0.832<br>0.808 | 37.988 | 0.879 |
| Perceived Similarity | - Similarity of Vision<br>- Selection Consistency<br>- Similarity of Recommended Information | 2.919 | 0.826<br>0.809<br>0.797 | 48.650 | 0.934 |
| Emotional Expectation Consciousness | - Mutual Tacit Behavior<br>- Mutual Expectation Consciousness<br>- Self-actualization Value | 2.173 | 0.808<br>0.808<br>0.796 | 36.224 | 0.909 |
| Parasocial Interaction | - Sharing Behavior Intention<br>- Information Utilization Recommendation<br>- Helping to Expand Relationships | 2.630 | 0.844<br>0.793<br>0.719 | 43.832 | 0.902 |
| Fad-like Behavior | - Similar Purchasing Process<br>- Recommendation-based Purchase Intention<br>- Imitation of Multiple Actions | 2.537 | 0.923<br>0.884<br>0.876 | 42.277 | 0.874 |

As a result of a further factor analysis using Varimax rotation, for the six items that described the community characteristics of Korean fashion/beauty creator media (Table 1), the two factors of "community scalability" (three items) and "community cohesion" (three items) were extracted with an eigenvalue of 1.000 or higher. The total variance explained by these two factors was 77.273%, and the Cronbach's α were all 0.879 or higher, indicating high reliability of the items.

In addition, through another factor analysis using Varimax rotation, for the six items that described the social exchange relationship characteristics in Korean fashion/beauty creator media, the two factors of "perceived similarity" (three items) and "emotional expectation consciousness" (three items) were extracted with an eigenvalue of 1.000 or higher (Table 1). The total variance explained by these two factors was 84.874%, and the Cronbach's α were all 0.909 or higher, indicating high reliability of the items.

Table 1 further shows the results of confirming the single dimension of the network effectiveness variables in Korean fashion/beauty creator media. Two factors were extracted with an eigenvalue of 1.000 or higher: "parasocial interaction" (three items) and "fad-like behavior" (three items); the factor loading was 0.719 or higher, and the reliability was high at 0.874.

### 4.3. Confirmatory Factor Analysis

The results of the confirmatory factor analysis are shown in Table 2. The standardization coefficients were all 0.6 or more, ensuring the concept validity, while the average variance extraction (AVE) was 0.5 or more in all instances, ensuring convergent validity. Furthermore, the reliability of each construct was over 0.7, ensuring internal consistency.

**Table 2.** Confirmatory factor analysis results.

| Measurement Item | Unstandardized Coefficient | Standardized Coefficient | SE | CR | Construct Reliability | AVE |
|---|---|---|---|---|---|---|
| Content Characteristics of Korean Fashion/Beauty Creator Media | | | | | | |
| Content Uniqueness | | | | | | |
| 1 | 1.000 | 0.810 | - | - | 0.834 | 0.719 |
| 2 | 0.975 | 0.800 | 0.039 | 13.785 | | |
| 3 | 0.947 | 0.747 | 0.032 | 11.600 | | |
| Content Continuity | | | | | | |
| 1 | 1.000 | 0.823 | - | - | 0.805 | 0.677 |
| 2 | 0.984 | 0.776 | 0.049 | 18.429 | | |
| 3 | 0.888 | 0.770 | 0.023 | 8.579 | | |
| Community Characteristics of Korean Fashion/Beauty Creator Media | | | | | | |
| Community Scalability | | | | | | |
| 1 | 1.000 | 0.766 | - | - | 0.908 | 0.759 |
| 2 | 0.992 | 0.764 | 0.032 | 7.919 | | |
| 3 | 0.987 | 0.752 | 0.026 | 6.593 | | |
| Community Cohesion | | | | | | |
| 1 | 1.000 | 0.799 | - | - | 0.859 | 0.739 |
| 2 | 0.978 | 0.761 | 0.031 | 11.345 | | |
| 3 | 0.926 | 0.753 | 0.022 | 8.039 | | |
| Social Exchange Relationship Characteristics | | | | | | |
| Perceived Similarity | | | | | | |
| 1 | 1.000 | 0.780 | - | - | 0.926 | 0.757 |
| 2 | 0.992 | 0.756 | 0.016 | 8.013 | | |
| 3 | 0.986 | 0.752 | 0.015 | 7.414 | | |
| Emotional Expectation Consciousness | | | | | | |
| 1 | 1.000 | 0.786 | - | - | 0.902 | 0.765 |
| 2 | 0.993 | 0.763 | 0.014 | 6.464 | | |
| 3 | 0.975 | 0.713 | 0.011 | 4.969 | | |
| Network Effectiveness | | | | | | |
| Parasocial Interaction | | | | | | |
| 1 | 1.000 | 0.800 | - | - | 0.899 | 0.727 |
| 2 | 0.917 | 0.779 | 0.025 | 10.583 | | |
| 3 | 0.776 | 0.711 | 0.013 | 5.410 | | |
| Fad-like Behavior | | | | | | |
| 1 | 1.000 | 0.785 | - | - | 0.851 | 0.721 |
| 2 | 0.968 | 0.770 | 0.032 | 12.644 | | |
| 3 | 0.954 | 0.764 | 0.019 | 7.472 | | |

Note. SE = standard error; CR = critical ratio; AVE = average variance extraction.

### 4.4. Model Fit Analysis

The fit and parameters of path analysis were estimated using the maximum likelihood method. First, the path analysis fitness index for the study of the network effectiveness of Korean fashion/beauty creator media was $X^2$ = 161.400 (df = 2, $p$ = 0.000), GFI = 0.946, AGFI = 0.938, RMR = 0.058, NFI = 0.927, CFI = 0.955, RMSEA = 0.034, indicating a satisfactory level. Thus, the relationships between the research concepts in the proposed model were jointly explained, as shown in Table 3.

**Table 3.** Model fitness measurement results.

| Model | Goodness-of-Fit Index | | | | | | | |
|---|---|---|---|---|---|---|---|---|
| | $X^2$ | df | *p*-Value | GFI | AGFI | RMR | NFI | CFI | RMSEA |
| Research Model | 161.400 | 2 | 0.000 | 0.946 | 0.938 | 0.058 | 0.927 | 0.955 | 0.034 |

Note. GFI = goodness of fit index; AGFI = adjusted goodness of fit index; RMR = root mean square residual; NFI = normed fit index; CFI = comparative fit index; RMSEA = root mean square error of approximation.

### 4.5. Research Hypothesis Testing

The results of testing the study's hypotheses on the structural influence relationship between the content and community characteristics of Korean fashion/beauty creator media, social exchange relationship characteristics, and network effectiveness are shown in Figure 2 and Table 4.

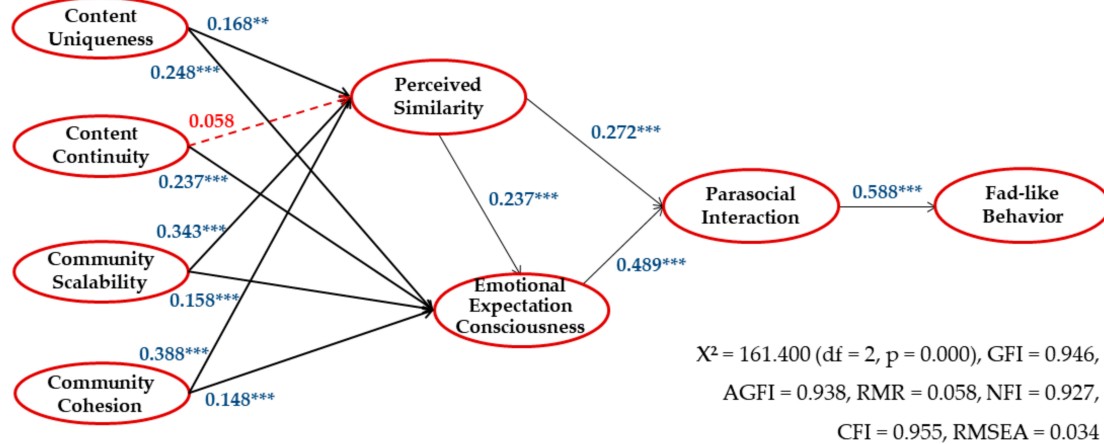

**Figure 2.** Research results model. (Note. * $p < 0.05$, ** $p < 0.01$, *** $p < 0.001$.)

**Table 4.** Research hypothesis testing results.

| Hypothesis | Path | | | Estimate | SE | CR | *p*-Value | Result |
|---|---|---|---|---|---|---|---|---|
| H1–1-1 | Content Uniqueness | → | Perceived Similarity | 0.168 | 0.052 | 3.265 | 0.001 ** | Accepted |
| H1–1-2 | Content Continuity | → | Perceived Similarity | 0.058 | 0.042 | 1.379 | 0.168 | Rejected |
| H1–2-1 | Content Uniqueness | → | Emotional Expectation Consciousness | 0.248 | 0.050 | 4.947 | 0.000 *** | Accepted |
| H1–2-2 | Content Continuity | → | Emotional Expectation Consciousness | 0.237 | 0.041 | 5.808 | 0.000 *** | Accepted |
| H2–1-1 | Community Scalability | → | Perceived Similarity | 0.343 | 0.042 | 8.247 | 0.000 *** | Accepted |
| H2–1-2 | Community Cohesion | → | Perceived Similarity | 0.388 | 0.048 | 8.138 | 0.000 *** | Accepted |
| H2–2-1 | Community Scalability | → | Emotional Expectation Consciousness | 0.158 | 0.040 | 3.907 | 0.000 *** | Accepted |
| H2–2-2 | Community Cohesion | → | Emotional Expectation Consciousness | 0.148 | 0.046 | 3.211 | 0.001 ** | Accepted |
| H3 | Perceived Similarity | → | Emotional Expectation Consciousness | 0.237 | 0.038 | 6.236 | 0.000 | Accepted |
| H4 | Perceived Similarity | → | Parasocial Interaction | 0.272 | 0.039 | 6.943 | 0.000 | Accepted |
| H5 | Emotional Expectation Consciousness | → | Parasocial Interaction | 0.489 | 0.037 | 13.331 | 0.000 | Accepted |
| H6 | Parasocial Interaction | → | Fad-like Behavior | 0.588 | 0.033 | 17.975 | 0.000 | Accepted |

Note: * $p < 0.05$, ** $p < 0.01$, *** $p < 0.001$.

Analyzing the path relationship between the content characteristics of Korean fashion/beauty creator media and the characteristics of social exchange relationships showed that content uniqueness had a significant influence on perceived similarity ($\beta = 0.168$, CR = 3.265, $p = 0.001$); however, content continuity did not ($\beta = 0.058$, CR = 1.379, $p = 0.168$). Content uniqueness also had a significant effect on emotional expectation consciousness ($\beta = 0.248$, CR = 4.947, $p = 0.000$), as did content continuity ($\beta = 0.237$, CR = 5.808, $p = 0.000$).

Second, the analysis of the path relationship between the community characteristics of Korean fashion/beauty creator media and social exchange relationship characteristics revealed that community scalability ($\beta = 0.343$, CR = 8.247, $p = 0.000$) and community cohesion ($\beta = 0.388$, CR = 8.138, $p = 0.000$) significantly affected perceived similarity. In addition, community scalability had a significant effect on emotional expectation consciousness ($\beta = 0.158$, CR = 3.907, $p = 0.000$), and community cohesion had a significant effect on emotional expectation consciousness ($\beta = 0.148$, CR = 3.211, $p = 0.001$).

Third, as a result of analyzing the path relationship between the social exchange relationship characteristics of Korean fashion/beauty creator media, it was found that perceived similarity had significant effects on emotional expectation consciousness ($\beta = 0.237$, CR = 6.236, $p = 0.000$) and parasocial interaction ($\beta = 0.272$, CR = 6.943, $p = 0.000$), and emotional expectation consciousness significantly affected parasocial interactions ($\beta = 0.489$, CR = 13.331, $p = 0.000$).

Finally, an analysis of the path relationship between parasocial interaction and fad-like behavior in Korean fashion/beauty creator media demonstrated that parasocial interaction had a significant influence on fad-like behavior ($\beta = 0.588$, CR = 17.975, $p = 0.000$).

Placing the results of our hypothesis testing in the context of related previous studies, the findings of the studies mentioned below can be interpreted in a manner similar to the results of H1 testing in the present study. In terms of the relationship between the content characteristics of Korean fashion/beauty creator media and social exchange relationship characteristics, it has been suggested that the differentiated characteristics of digital SNS content (interactivity, asynchronisity, mobility, multimedia realization, terminal expandability, etc.) change the environmental usefulness of the audience while ensuring mutual interactions and interactive information distribution [43]. In one-person media, the fun factor of the content and the honesty and intimacy of the creators are the characteristics that influence users to feel positively toward the content, providing an opportunity for new social network formation, participation, and empathy [48]. It has also been argued that even an international sense of solidarity can be spread according to the quantitative and qualitative delivery message of content [42].

The findings from the testing of this study's second hypothesis support those of the studies mentioned below. In terms of the relationship between the community characteristics of Korean fashion/beauty creator media and social exchange relationships, identity and usefulness, exchange with friends, searching for information about friends, leisure, personal network management, escape from reality, participation, and escape from loneliness are the main motivations for use in community-based social media [49]. For members to gain satisfaction through the formation of trust between community operators and members, it is necessary to provide economic or non-economic activity compensation to motivate community participation [43]. Further, knowledge sharing in the online community is developed through a sense of future expectation consciousness based on reciprocity ("mutual benefit") to prevent "free rides" [29].

Third, the findings of the studies mentioned below support the results of H3, H4, H5, and H6 in the present study. In terms of the relationship between social exchange relationship factors and network effectiveness in Korean fashion/beauty creator media, the community structure is shaped through network characteristics based on the links between the community's commonality, unified consciousness, and similarity [50]. Members belonging to subnetworks, connected by strong ties, value the embeddedness formed through these ties from the viewpoint of sharing characteristics that are distinct from members of other subnetworks [45]. Moreover, social exchange has a significant effect on members' organizational citizenship behavior [44], and social interaction increases the level of viewers' satisfaction by increasing the intimacy between viewers and creators [51]. Furthermore, social interactivity can act as social capital that is positively connected to the sense of community [46]. The interactions that appear as emotional connections between SNS users influence the attitudes toward specific products and purchase intentions within the SNS group [52].

## 5. Conclusions

Owing to the increasing number of platforms through which international consumers can access various Korean fashion/beauty video content, the pattern of brand and product recognition is changing. Therefore, in terms of marketing, the issue of how international consumers, marketers, and creators should collaborate through content and community use is becoming more important in the activities of creator media. In this context, the present study verified the structural impact relationships between the content and community characteristics of Korean creator media, the social exchange relationship characteristics between this media and its participants, and the network effectiveness of the participant group.

The study's theoretical implications are as follows. First, we presented the content and community characteristics of Korean fashion/beauty creator media as factors for each dimension and evaluated the characteristics of these creator media in the cognitive,

emotional, and behavioral stages to assess media communication effects. This process offers a theoretical framework for hierarchical audience responses and formation roles as important antecedent variables and process approaches in relational marketing and media acceptance research with international consumers. Second, it was verified that the content and community influence of Korean fashion/beauty creator media is necessary in the process of forming social exchange relationships and network effectiveness among international consumers. In particular, by linking social exchange relationships with the issues of similarity and expectation from a network perspective, the study expanded the scope and target of academic research on social exchange relationships between Korean fashion/beauty creator media and international consumers. Third, the study suggests a direction for future research that can approach the effectiveness of networks of international consumers of Korean fashion/beauty who are exposed to Korean fashion/beauty creator media from a behavioral and social psychological perspective. In addition, a structural model that highlighted fad-like behavior among members of an interest group for Korean fashion/beauty products through this influence relationship was presented. This is expected to enable the development of a theoretical model that further explains attitudes toward Korean fashion/beauty products and purchase intentions when integrated with the existing Hallyu culture, as well as studies related to the acceptance of Korean fashion/beauty products by international consumers.

Based on the study's results, the following marketing implications are proposed. First, to increase the perceived similarity between Korean fashion/beauty creator media and participants, it is necessary to tailor Korean fashion/beauty creator media content to foreigners and offer customized services (e.g., English subtitles). Overall trust and belief should also be increased by providing professional information and increasing the uniqueness of content through continuous new updates. It is also necessary to increase curiosity by using influencers for network continuity, providing additional product purchase methods by linking additional genres and brands, and increasing community scalability through segmentation (grouping) and the internationalization of communities. The bond between Korean fashion/beauty creator media services and consumers should be strengthened by prioritizing public interest and ethics within the network, enabling the use of tools for network communities, and increasing community cohesion to enable the formation of cooperative partnerships and to revitalize events for network cohesion.

Second, to increase the emotional expectation consciousness between Korean fashion/beauty creator media and its participants, it is necessary to enhance content uniqueness, community scalability, community cohesion, and the original storyline of the Korean fashion/beauty creator media content. It is also necessary to diversify the content's subject matter, form, creativity, and differentiation and increase content continuity by utilizing trends and the feeling of Synesthesia. Perceived similarity should be increased by enhancing the experience of similarity among participants through the information obtained from Korean fashion/beauty creator media, confirming which information is of interest to participants, and sharing the opinions of participants on posted information.

Third, to increase the parasocial interaction among viewers of Korean fashion/beauty creator media, the value of self-actualization should be increased through the emotional expectation consciousness and immersion in Korean fashion/beauty creator media, as well as perceived similarity. Emotional expectation consciousness should also be encouraged through the formation of a tacit relationship between mutual help and stimulation from the media.

Fourth, to increase the similar purchase process, intention to purchase recommended products, and fad-like behavior of Korean fashion/beauty creator media viewers, it is essential to directly request the necessary information from this media. In addition, parasocial interactions should be enhanced to enable the expansion of emotional relationships through the media connections of personal content sharing, use, and participation.

In this study, the sustainable social exchange relationship and network effectiveness formed in Korean fashion/beauty creator media were measured by surveying international

consumers who come into contact with Hallyu popular culture through such media, recognize Korean fashion/beauty within it, and subsequently purchase Korean fashion/beauty products. However, in this study, there is a limit to the identification of various specific strategic elements by category in the fashion/beauty field. For this purpose, detailed research on content and community characteristics that can create and seek differentiated values including characteristics of each field should be continuously conducted. Further, the scope of the present study should be further expanded by examining the suitability and validity of more diverse evaluation factors in the future. In follow-up research, it will be necessary to examine the detailed marketing strategy factors that can lead to sustainable value and co-creation with community support, develop various methods for improving creator content, and identify branded content and media mix strategies that incorporate brands within content. This will allow the development of practical data and strategic directions for the future of creator media and the fashion/beauty field.

**Author Contributions:** Y.N. and S.K. wrote the paper and designed the survey. Y.N. contributed to the writing of the paper and conducted the survey and data analysis. S.K. provided research insights and contributed to the majority of the writing and editing of the paper. H.J. assisted in survey implementation and data analysis. All authors have read and agreed to the published version of the manuscript.

**Funding:** This work was supported by the Ministry of Education of the Republic of Korea and the National Research Foundation of Korea (NRF-2019S1A5B5A07105152).

**Data Availability Statement:** Data is not publicly available, though the data may be made available on reasonable request from the first author.

**Conflicts of Interest:** The authors declare no conflict of interest.

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
