# Peer review of "A Study on the Network Effectiveness of Sustainable K-Fashion and Beauty Creator Media (Social Media) in the Digital Era"

_sustainability, doi:10.3390/su13168758_

Round 1

Reviewer 1 Report

The research article presented for review A Study on the Network Effectiveness of Sustainable K-Fashion and Beauty Creator Media (Social Media) in the Digital Era: Focusing on Content and Community Characteristics consists of seven basic parts – abstract, introduction, theoretical background, review of applied research method and analysis of obtained research results, conclusions resulting from research results with presentation of the study’s implications and a list of literature used for the implementation article.

The paper is in line with actual and contemporary approach to the responsible management and sustainable paradigm. The study is very interesting, it can be improved but in my opinion it meets the minimum quality to be accepted for publication.

I would like to get clarifications from the authors what is precisely meant by Hallyu phenomenon and what are its components.

I would recommend to authors to improve clarity of the paper title – it’s to long and needed to be shortened.

Its scientific value could also be enriched by supplementing the references and carrying out a solid review of the literature on the subject.

I hope you find my comments helpful. Best of luck with the revisions.

The abstract doesn’t clearly indicate the structure of the work. It points out and justifies to the small extent the purpose of the study. The weakness of the abstract is that it didn’t highlight what added value and contribution the article brings to the literature of the subject – network analysis and creator media conditions of functioning.

The literature review should be more extensive – it should carefully emphasize contemporary works in the field of methodology, which are published in recognized scientific journals, and indicate current research directions.

A valuable element of the article is its conceptual order, correctly and unambiguously formulated hypotheses and advanced research procedure.

Unfortunately, the article in the conclusions section doesn’t indicate limitations of the paper as well as further research directions and other works that undertake similar research issues, which could constitute the basis for comparative analyses.

Author Response

Cover Letter for Revisions Made in the Manuscript

- Based on the assistant editor notes for minor revisions -

(received on 28 July 2021)

We appreciate the assistant editor notes for minor revisions of our research paper for publication. In our effort of improving the paper and meeting the “within 5 days revised paper submission” requirement, we carefully completed the revision and editing work as suggested by the reviewer(s). We have carefully examined the comments and made our best effort in providing appropriate revisions to finalize the paper. We used the file which is downloaded from the system for the revision work. Please note that only the current revisions made based on the reviewer comments are indicated by the “track changes memo” in the paper. Overall, the writing of the paper is checked and modified by professional editing service to improve English for better readability and clarity.

The specific contents of the revision work are addressed below. We greatly appreciate your keen interest on our paper and we look forward to hearing from you again with a positive final decision. Thank you.

Sincerely yours,

Sungmin Kang

Professor of MIS

Chung-Ang University

Title: A Study on the Network Effectiveness of Sustainable K-Fashion and Beauty Creator Media (Social Media) in the Digital Era

Reviewer #1: Comments and Suggestions for Authors

Author Responses to Reviewer Comments

The paper is in line with actual and contemporary approach to the responsible management and sustainable paradigm. The study is very interesting, it can be improved but in my opinion it meets the minimum quality to be accepted for publication.

We appreciate your positive comments and overall compliment. We addressed the comments made and made relevant revisions as they help to improve the overall quality of the paper.

I would like to get clarifications from the authors what is precisely meant by Hallyu phenomenon and what are its components.

We have elaborated on Hallyu by providing additional description. Hallyu is discussed in detail in the introduction section.

I would recommend to authors to improve clarity of the paper title – it’s too long and needed to be shortened.

The title of the paper is shortened for its clarity.

Its scientific value could also be enriched by supplementing the references and carrying out a solid review of the literature on the subject.

We tried to provide additional references for enriching the review of literature on the subject and relevant findings.

I hope you find my comments helpful. Best of luck with the revisions.

Your comments are appreciated and we did our best to conduct the relevant revisions. Thank you!

The abstract doesn’t clearly indicate the structure of the work. It points out and justifies to the small extent the purpose of the study. The weakness of the abstract is that it didn’t highlight what added value and contribution the article brings to the literature of the subject – network analysis and creator media conditions of functioning.

We have discussed the key components in the abstract. Considering the 200 words limit, it was difficult to mention everything and more. In any case, we tried to discussed the value and contribution of the article by adding new information in the abstract.

The literature review should be more extensive – it should carefully emphasize contemporary works in the field of methodology, which are published in recognized scientific journals, and indicate current research directions.

Literature review on relevant topics of the paper are provided. Research on one-person media is a relatively more recent phenomenon and not much research is conducted in this area. Thus, current research direction is indicated with more elaboration in the introduction section. Additional issues are further referenced in the paper.

A valuable element of the article is its conceptual order, correctly and unambiguously formulated hypotheses and advanced research procedure.

We thank you for your compliments on paper quality. We conducted this study based on well planned research methodology.

Unfortunately, the article in the conclusions section doesn’t indicate limitations of the paper as well as further research directions and other works that undertake similar research issues, which could constitute the basis for comparative analyses.

We further discussed the limitation of the paper, suggesting the future research directions as well in the conclusion section.

Reviewer 2 Report

Thank you for giving me the opportunity to review the paper. It's an interesting topic and I read it well. However, I'd like to give the author(s) few comments.

  1. In Abstract, do not end by listing results, but also suggest implications.

  1. The authors conducted reliability and value analysis and CFA for each variable, but I think it would be better to do it all at once.

  1. Please present Table 1-4, and Table 5 comprehensively and with increased visibility.

Author Response

Cover Letter for Revisions Made in the Manuscript

- Based on the assistant editor notes for minor revisions -

(received on 28 July 2021)

We appreciate the assistant editor notes for minor revisions of our research paper for publication. In our effort of improving the paper and meeting the “within 5 days revised paper submission” requirement, we carefully completed the revision and editing work as suggested by the reviewer(s). We have carefully examined the comments and made our best effort in providing appropriate revisions to finalize the paper. We used the file which is downloaded from the system for the revision work. Please note that only the current revisions made based on the reviewer comments are indicated by the “track changes memo” in the paper. Overall, the writing of the paper is checked and modified by professional editing service to improve English for better readability and clarity.

The specific contents of the revision work are addressed below. We greatly appreciate your keen interest on our paper and we look forward to hearing from you again with a positive final decision. Thank you.

Sincerely yours,

Sungmin Kang

Professor of MIS

Chung-Ang University

Title: A Study on the Network Effectiveness of Sustainable K-Fashion and Beauty Creator Media (Social Media) in the Digital Era

Reviewer #2: Comments and Suggestions for Authors

Author Responses to Reviewer Comments

Thank you for giving me the opportunity to review the paper. It's an interesting topic and I read it well. However, I'd like to give the author(s) few comments.

We are glad to know that paper review was enjoyable. Thank you for your positive comments, and your advices are well addressed in the paper revision for improving the paper.

In Abstract, do not end by listing results, but also suggest implications.

We have discussed the key components in the abstract. Considering the 200 words limit, it was difficult to mention everything and more. In any case, we tried to discussed the value and contribution of the article by adding new information in the abstract.

The authors conducted reliability and value analysis and CFA for each variable, but I think it would be better to do it all at once.

The relevant analyses are conducted all at once for the research variables. Thank you for the suggestion.

Please present Table 1-4, and Table 5 comprehensively and with increased visibility.

Table 1-4 are comprehensively combined for increased visibility. Table 5 is already formatted by the Sustainability as it meets the format requirement of Sustainability Journal.

Reviewer 3 Report

This paper was a nice read. I liked very much the detailed methodological description and well made explanation of the research design. Statistical tests seem to be ok and they are well defined with high explanation levels. The content topic is very innovative and I have not seen anything concerning this topic before. Therefore, the paper has novelty in terms of idea and execution. The paper also addresses future topics and questions that this study does not provide answers to.  It’s well done research.

Author Response

Cover Letter for Revisions Made in the Manuscript

- Based on the assistant editor notes for minor revisions -

(received on 28 July 2021)

We appreciate the assistant editor notes for minor revisions of our research paper for publication. In our effort of improving the paper and meeting the “within 5 days revised paper submission” requirement, we carefully completed the revision and editing work as suggested by the reviewer(s). We have carefully examined the comments and made our best effort in providing appropriate revisions to finalize the paper. We used the file which is downloaded from the system for the revision work. Please note that only the current revisions made based on the reviewer comments are indicated by the “track changes memo” in the paper. Overall, the writing of the paper is checked and modified by professional editing service to improve English for better readability and clarity.

The specific contents of the revision work are addressed below. We greatly appreciate your keen interest on our paper and we look forward to hearing from you again with a positive final decision. Thank you.

Sincerely yours,

Sungmin Kang

Professor of MIS

Chung-Ang University

Title: A Study on the Network Effectiveness of Sustainable K-Fashion and Beauty Creator Media (Social Media) in the Digital Era

Reviewer #3: Comments and Suggestions for Authors

Author Responses to Reviewer Comments

This paper was a nice read. I liked very much the detailed methodological description and well-made explanation of the research design. Statistical tests seem to be ok and they are well defined with high explanation levels. The content topic is very innovative and I have not seen anything concerning this topic before. Therefore, the paper has novelty in terms of idea and execution. The paper also addresses future topics and questions that this study does not provide answers to. All in all, I think you could publish the paper as it is. It’s well done research.

We are honored to see your compliments in the reviewer comment. We definitely did our best to conduct the research and write the research paper through verification procedures and editing service. We tried to apply sound research methodology. We thank you for understanding our effort and dedication in conducting this research for paper publication. We also look forward to conducting more novel research in the future with your encouragement and support.
